# A Review on Principles, Theories and Materials for Self Sensing Concrete for Structural Applications

**DOI:** 10.3390/ma15113831

**Published:** 2022-05-27

**Authors:** Kousalya Ramachandran, Ponmalar Vijayan, Gunasekaran Murali, Nikolai Ivanovich Vatin

**Affiliations:** 1Department of Civil Engineering, College of Engineering, Guindy, Anna University, Chennai 600 025, India; donaramachandran96@gmail.com; 2Peter the Great St. Petersburg Polytechnic University, 195251 St. Petersburg, Russia; murali_22984@yahoo.com (G.M.); vatin@mail.ru (N.I.V.)

**Keywords:** self-sensing, piezoresistive, piezopermittive, electrical resistance, percolation threshold, tunnelling effect, gauge factor

## Abstract

Self-sensing concrete is a smart material known for its cost-effectiveness in structural health-monitoring areas, which converts the external stimuli into a stress/strain sensing parameter. Self-sensing material has excellent mechanical and electrical properties that allow it to act as a multifunctional agent satisfying both the strength and structural health-monitoring parameters. The main objective of this review is to understand the theories and principles behind the self-sensing practices. Many review papers have focused on the different types of materials and practices that rely on self-sensing technology, and only a few articles have discussed the theories involved. Understanding the mechanism and the theories behind the conduction mechanism is necessary. This review paper provides an overview of self-sensing concrete, including the principles such as piezoresistivity and piezopermittivity; the tunnelling effect, percolation threshold, and electrical circuit theories; the materials used and methods adopted; and the sensing parameters. The paper concludes with an outline of the application of self-sensing concrete and future recommendations, thus providing a better understanding of implementing the self-sensing technique in construction.

## 1. Introduction

Concrete is an excellent building material developed 200 years ago by the Roman Empire. It is known for its high strength, affordability, aesthetic and durable nature. It has applications in several forms, from simple to complex structures such as nuclear power plants, dams, tunnels, residential buildings, roads, and pavement [1]. These building materials are exposed to heavy loads and aggressive environments, resulting in aging, cracking, temperature variation, deterioration, sulphate attack, corrosion, etc. [2]. Therefore, the structure should be designed to satisfy its function in a severe and aggressive environment. In that case, the material and design are the main factors considered during the preconstruction and construction phases. In addition, at the time of serviceability, the structure has to be periodically monitored to detect the cracks and other damaging parameters and should be maintained accordingly; this technique is termed structural health monitoring (SHM). SHM is a technique of examining a structure’s performance by periodically providing information on stress, displacement and cracks as well as other information related to the real-time condition of the structure through various sensors. SHM helps reduce the cost of repair and maintenance throughout the life span. The technique consists of onboard sensors for data acquisition and a central processing unit for monitoring purposes [3,4,5].

In the 19th century, many sensors were developed to monitor the performance of buildings. The strain gauge was the first sensing device introduced to monitor displacement and stress, followed by accelerometer and extensometer sensors. Later, due to its complication in instrumentation set up and low sensitive behaviour of conventional sensors, smart sensors like piezoelectric sensors, fibre optical grating, wireless sensors and shape memory alloys were introduced [6,7,8,9,10,11,12,13,14,15,16]. The commonly used conventional sensors are shown in Table 1. However, attached and embedded sensors have drawbacks such as poor durability, short span, low sensitivity, and low compatibility [17].

Research and development have recently introduced intrinsic self-sensing concrete to overcome the above issues by satisfying both the strength parameter and the structural health monitoring purposes. Self-sensing is a method based on the principle of conversion of mechanical or chemical parameters into an electrical sensing output. Many researchers have implemented this method in several applications, such as strain and stress detecting [18], traffic monitoring [19], corrosion monitoring [20], earthquake monitoring [7] and crack detection. Figure 1 presents various applications of self-sensing techniques for structural and health monitoring purposes.

The past works of literature related to self-sensing technologies are graphically represented in Figure 2. This study concentrates on the electromechanical and electrochemical mechanisms that self-sensing concrete relies on. The theories, materials, sensitivity and influencing parameters are discussed. This review paper discusses the general applications of self-sensing concrete, the drawbacks, and the research that must be focused on in the future.

## 2. Sensing Mechanism

To monitor the real-time stress and strain in concrete structures, researchers have developed many health-monitoring techniques, among which self-sensing is one of the emerging ones. Self-sensing is a technique that responds to external factors such as loading conditions, environmental variations, and temperature variations by converting them into electrical output properties [20]. This conversion can be obtained by adding conductive fillers to the matrix material, applying an electric field, or connecting the composite with equivalent circuit models. Although the initial cost is high compared with traditional sensors, the later maintenance and repair work can be neglected and is comparatively economical. Various attributes can determine the sensing nature of the material to measure the stress or strain, including electromechanical, electromagnetic, electrochemical, dielectric, magnetic, optical, etc.

Among these attributes, the electromechanical and electrochemical mechanisms are research focuses due to their fast response to external conditions [7]. This review discusses a detailed study of the above two mechanisms.

## 3. Electromechanical Mechanism

Electromechanics is a principle in which the sensing nature is determined through electrical properties applying different loading conditions. The electromechanical sensing mechanism discussions are analysed based on three main perspectives: piezoresistivity, piezopermittivity and piezoelectric performances. Piezoresistivityand piezopermittivity deal with self-sensing techniques, whereas piezoelectric materials is not suitable for self-sensing techniques. In this review, a detailed review of piezoresistivity and piezopermittivity are discussed. Table 2 presents different techniques involved in electromechanical principles.

### 3.1. Piezoresistive Performance

Piezoresistivity is defined as the resistivity of material that changes with strain, i.e., the conversion of external load to electrical resistivity, or, in other words, change in resistivity on the application of an external loading condition [20,21]. It is the most widely used method in self-sensing concrete as the sensing parameters provide more accurate results than other self-sensing techniques; adding conducting filler to the matrix improves both the strength and sensing parameters. The change in resistivity is due to the change in dimensions when the material is subjected to strain or stress [22,23]. Figure 3a represents the instrumental set up for the piezoresistive method, and Figure 3b shows piezoresistive behaviour of the specimen when subjected to compression load.

Electrical behaviour in the composite is due to the presence of conductive fillers. For intrinsic self-sensing material, the electrically conducting fillers such as fibres and particles are fused with cement or polymer-based matrix (cement mortar, cement paste or concrete, polymer). The electrical conductivity of the commonly used functional fillers is shown in Table 3 [24,25,26].

#### 3.1.1. Materials


**Conductive material:**


The conductive functional fillers are responsible for piezoresistive behaviour, capable of sensing stress, strain, deflection, cracks, humidity, and temperature by forming a conductive path along with the matrix material [27]. The functional fillers include fibrous and powdered materials (micro and nano) such as steel fibres, nickel powder, graphite nano-derivatives like graphene powder, reduced graphene oxide and carbon micro- and nano-functional fillers such as carbon black, carbon fibres (short and continuous), carbon nanotubes and carbon nanofibers [28,29,30,31]. The hybrid fillers, such as graphene-coated carbon fibres and nickel-coated fibre materials, can also provide excellent sensing properties to the composite material.


**Non-conductive material:**


The matrix element refers to an insulating or semi-insulating material with electrical resistivity ranging from 10 to 10^2^ Ω·cm. The role of matrix material is to bind the fillers together to form a bulk composite. For structural self-monitoring purposes, the generalized matrix material is cement-based composites, including concrete, mortar and cement paste. Other material composites like geopolymer and alkali-activated composites are currently under research.

The sensing ability of the matrix material is poor, but it directly depends on mechanical behaviour such as stress and strain, which depends on electrical conductivity [32,33].

**Table 3 materials-15-03831-t003:** Types of fillers used in self-sensing and their properties.

Conductive Material	Geometric Shape	Tensile Strength (GPa)	Elastic Modulus (GPa)	Aspect Ratio	Density (kg/m^3^)
Steel fibres (straight) [25]	Fibre (Micro filler)	500	200	97.5	7850
Steel fibres (twisted) [26]	Fibre (Micro filler)	2428	200	100	7900
Carbon nano fibre [27,28]	Particle (Micro filler)	4900	230	100–500	1000
CNT [29]	Particle (Nano filler)	11	300–1000	~1000	50–150
Nano graphene platelets [30,31]	Particle (Nano filler)	5000	1000–2000	50–300	1800
Carbon black [31,34]	Particle (Nano filler)	2000–2400	-	120	1800–2100

#### 3.1.2. Sensing Theories

Piezoresistivity depends on ionic and electron transfer mechanisms. Ionic transfer relates to the ionic motion in the composite material at saturation time: When the pores in the structure fill with water or another moist substance, the ionic species (Na^+^, Ca^+^, K^+^, OH^−^, SO_4_^2−^) dissolve from their solid state [35], which decreases resistivity. The electron transfer occurs in a dry state where the transfer of ions is difficult. Therefore, the conductivity occurs by tunnelling current. The tunnelling current is assumed to be developed by two theories, the percolation theory and the field emission effect [31,36,37,38,39,40,41,42,43].


**Percolation theory**


The percolation theory is used to analyse the physical properties of a heterogeneous composite and can be explained by the formation of conductive paths [44]. The concept lies in the fact that when the conductive particles come in contact with each other or the volume of the fraction of fillers approaches a critical value (i.e., percolation threshold $\varphi_{c})$, a continuous network is extended throughout the system [44,45,46,47,48].

The conduction process undergoes three conditions depending on the volume of the fraction of functional fillers. Figure 4 shows a graph divided into three zones with the volume of fractional fillers (φ) along the X-axis and electrical resistivity along theY-axis. Zone A indicates the insulating region, where the ionic conduction takes place due to the hydration process in the cement matrix and the filler concentration (φ) is less than the critical filler content ($\varphi_{c})$, that is, $\varphi <\varphi_{c}.$ In Zone B, on the application of external stimuli, the composite material changes from an insulating medium to a conducting medium. The filler materials come in contact with each other, resulting in a percolation process $\left(\varphi =\varphi_{c}\right)$. Guessero etal. studied percolation theory in which the conductivity takes place due to the electron transfer between the fillers, which results in a tunnelling effect, as shown in Figure 5. Zone C indicates the conducting region, where the filler concentration exceeds the critical filler concentration $\;(\varphi >\varphi_{c})$. The percolation threshold $(\varphi_{c})\;$ depends on many factors such as filler concentration, filler size and filler orientation, as shown in Table 3. The electrical conductivity is determined by the power law [47],
(1)$$\sigma =\sigma_{o}{\left(\varphi_{f}-\varphi_{c}\right)}^{s}$$
where σ is the electrical conductivity of the composite, $\sigma_{o}$ is the electrical conductivity of the filler, $\varphi_{f}$ is the volume fraction of the filler, $\varphi_{c}$ is the percolation threshold and *s* is the conductivity exponent. The percolation threshold indicates the point of a material’s transformation from an insulator to a conductor.

##### Parameters Influencing Percolation Threshold

Filler Geometry:

The shape of the filler material can influence the percolation value. Particles with a high surface area and high aspect ratio can form a conductive network below the percolation threshold. The fractional volumes of filler for different filler geometries based on the interparticle distance between the two conductive fillers are provided in Table 4 [45].

Filler and matrix properties:

The properties of the filler and the matrix condition play an important role in the formation of the conducting network. The properties near the percolation zone can be given by the following formula:$$Properties\;\alpha \left|f-f_{c}\right|$$
where  f is the volume fraction of the minor phase and $f_{c}$ refers to the percolation threshold. For a homogeneous composite with randomly distributed fillers,$\;f_{c}$ is approximately 0.16, which is called a Sher-Zallen invariant [49,50].

The filler material plays an important role in the formation of the network. The percolation threshold changes for different shapes of filler and depends on the homogeneity of the composite member. Based on the nature of the composite and the shape of the filler material, the percolation threshold can be determined as given in Table 4, in which R_1_ represents the particle size of the composite material and R_2_ represents the particle size of the conductive filler material [49].

Filler Concentration:

The concentration of filler in the composite plays a crucial role in forming the network and percolation threshold value. The conductivity of the sensing material decreases when the filler concentration is less than the critical filler concentration; the transformation of phase from insulating to conducting medium occurs when the filler concentration becomes equal to the critical filler concentration. The conductivity of the sensing material increases with an increase in filler concentration, but when it exceeds the critical filler value, a cluster of networks is formed nearby thus making the material unstable at the time of measurement [50].

The percolation threshold varies for different fibre materials concerning the corresponding matrix; Table 5 shows the percolation thresholds for different fibres with their percentages.

The percolation theory assumes that the piezoresistive behaviour, that is, the change in resistivity near $\varphi_{c}$, is due to the conductivity of the conductive fillers that are in contact with each other [51,52]. The theory is not applicable to discontinuous conductive particles.


**Field emission effect:**


Field emission theory gives a better explanation for the non-contacted filler particles. The theory states that a potential barrier forms between the non-contacted conductive filler material that develops conductivity when the conductive particles overlap or the gap between the particles ranges in nanometre distances [57]. This conductivity phenomenon is due to the tunnelling effect. Simmons derived an equation for the tunnelling effect [58]:(2)$$J=\left[\frac{3{\left(2m\varphi \right)}^{\frac{1}{2}}}{2S}\right]{\left(\frac{e}{h}\right)}^{2}V$$
where *J* is the current density, φ and *S* are the gap barrier and gap width, *m* is the electron mass, *e* refers to the charge of a single unit and *h* is the Planck constant. However, the tunnelling effect due to the field emission effect has limited significance to the tunnelling effect that occurs due to contact conductivity; it does, however, significantly enhance the piezoresistive properties. This is due to the particle gap, which is too large for field emission theory, in which the conductivity becomes difficult. The filler gap can be adjusted by increasing the filler concentration [59,60].

#### 3.1.3. Sensing Techniques and Measuring Parameters


**Sensing technique**


Sensing in concrete materials takes place using either two- or four-probe electrode configuration methods, as shown in Figure 6. Even though the two-probe method, the simplest and most commonly used approach in research, works to determine the material’s resistance, the four-probe method gives a better result by eliminating the contact resistance between the electrodes and the composite material [61,62]. In addition to the measurement of resistance, the electrode material, which acts as a bridge between the cement composite and the measuring elements, plays a critical role. The electrode should have low electrical resistance and stable electrical conductivity. Metals like copper, stainless steel, silver and aluminium are used as electrodes in the form of a metal plate with or without a hole, metal foil, mesh, a bar and copper wire wrapped with conductive paints such as silver, copper and carbon black [61,62,63,64,65].

To measure the electrical behaviour of the composite, two types of current modes are used: direct current (DC) and alternating current (AC). The direct current test is said to be the simplest method, but the current does not travel long distances, and it can lead to the movement of ions, resulting in electrical polarization in the composite. Due to the electrical polarization, it is difficult to measure the electrical resistance; therefore, to overcome this problem, DC voltage is applied over the composite before the time of loading so that the polarization is complete at the time of measurement. Another approach to overcoming this problem is using alternating current (AC), where the polarization still occurs, but it can be altered by increasing the frequency range and lowering the amplitude of the AC voltage [66,67,68,69]. Table 6 shows the electrical behaviour for different matrix and filler materials through different methods. Measuring piezoresistivity is complicated due to the presence of an electrode as a medium; for this reason, implementing this method in practical structural applications is complicated.


**Measuring parameters:**


The performance of piezoresistivity can be determined using a cube or prism under different loading conditions in anelastic regime, a plastic deformation region and a failure condition. The electrical nature of the material plays a dominant role in measuring the sensing parameters. The electrical resistance is determined using Ohm’s law based on two different conditions. Under the loading condition, the voltage (V) and current (I) are measured from the ammeter, from which the resistance of the material is determined [68,69,70]. The Table 7 shows the resistance formula for different circuit system. 

The sensing behaviour can be determined by sensitivity parameters such as a fractional change in electrical resistivity or a force sensitivity coefficient, stress sensitivity coefficient or strain sensitivity coefficient (also called a gauge factor). The sensitivity properties for different filler materials are listed in Table 8 [71,72,73,74,75,76,77,78,79].

The resistivity offered against the electrical conductivity (*ρ*) and the difference in resistivity (Δ*ρ*) can be determined using the following equations [70,71]:(3)$$\rho =R\cdot \left(\frac{A}{l}\right)$$
where *ρ* refers to the electrical resistivity (Ω·cm), *A* is an area at the cross-section of the specimen (cm^2^), *l* refers to the distance between the consecutive electrodes (cm) and *R* refers to the resistance of the material. From Equation (3), the fractional change in resistivity can be obtained as
(4)$$\frac{\delta R}{R}=\frac{\delta \rho }{\rho }+\left(\frac{\delta l}{l}\right)\left(1+2µ\right)$$
where δR/R = the fractional change in resistance, δρ/ρ = the change in resistivity, δl/l= longitudinal strain and µ = Poisson’s ratio. The value of fractional resistivity varies based on the loading conditions, and it is adopted for quantitatively evaluating the self-sensing capacity of a composite. The gauge factor of the sensing material is stated as a fractional change in resistance to strain (per unit); this parameter is used to quantitatively evaluate the feasibility of a composite as a sensor, and it is given by
(5)$$GF=\frac{\left(\frac{\delta R}{R}\right)}{\frac{\delta l}{l}}$$

Based on Equation (3), both changes in resistivity and change in strain can result in a change in resistance, so three conditions are adopted for determining the sensitivity, as shown in Table 8 [32].

Table 9 shows the sensitivity parameters for different fibre proportions on different cementitious composites. As the loading condition and volume of fibre proportion increase, the sensing parameter varies. The higher the loading rate, the greater the FCR and the lower the resistivity [80]. The gauge factor increases with the increase in loading rate and fibre proportion, and a higher gauge factor provides higher sensitivity. The table displays that carbon nanotube and carbon fibre show greater sensitivity in the cementitious composite.

### 3.2. Piezopermittivity

The relative electrical permittivity describes the dielectric behaviour of a material. It involves capacitance-based measurement. The capacitance in a composite is due to the polarization resulting from the movement of charge carriers when the material is subjected to an electric field. When the electric field is applied, the ions in the medium are repelled, which causes a dipolar effect, resulting in polarization. The effect of strain due to the application of loading conditions on permittivity is referred to as piezopermittivity. This type of technology is suitable for both new and existing structures. Permittivity is one of the main factors governing the sensing behaviour of the composite material [41,91,92].

The piezopermittivity technique is more advantageous than piezoresistivity because the electrodes need not be in intimate contact with the matrix composite and a conductive filler is not necessarily required, making the measuring technique easier and more economical. Many research works have been focused on piezoresistivity, but relatively little research has been reported on piezopermittivity.

The addition of conductive fillers to a matrix material results in an increase in permittivity due to the polarization effect. High permittivity is attractive for E.M.I. shielding, and low values are adopted for electromagnetic transparency. Polarization occurs in both conductive and non-conductive materials, and the mechanism is different in the two mediums. In a conductive medium, polarization occurs due to the interaction between the charged carrier particles. In non-conductive materials, the heterogeneous nature of the material results in polarization [36,93,94,95,96,97,98].

#### 3.2.1. Materials

In capacitance-based sensing materials, functional fillers are not required for determining the sensing characteristics. The materials with high permittivity are used as capacitors in piezopermittivity measurement. Matrix material or non-conductive material such as cement or ceramic has high permittivity with relatively low electrical conductivity.

Cement is a dielectric medium; ionic conduction takes place in the presence of moisture, and with the addition of conductive filler, electrical conduction takes place [94,95]. With the addition of nanofiller to the matrix material, the relative permittivity decreases as it occupies the filler space by limiting the polarization effect. With the addition of carbon fibres or microparticles to the matrix, the relative permittivity increases [95,96,97,98,99]. The permittivity and resistivity values for different filler materials are provided in Table 10 [95,96,97,98,99,100].

#### 3.2.2. Sensing Theory


**Sensing method:**


Permittivity is determined by measuring the capacitance of the composite material. The experimental method consists of a specimen sandwiched by a dielectric film and the electrode. The L.C.R. meter is not suitable for measuring the capacitance of a low resistive material; therefore, an electrically insulated sheet (dielectric film) is positioned between the electrode and specimen. The commonly used electrodes are copper wire or copper rods. The electrodes are bonded to the upper and lower surfaces of the specimen using double-sided adhesive tape. This adhesive tape acts as a dielectric film [101,102,103,104,105,106]. The specimen setup is shown in Figure 7.

The tests need to be performed for a specimen with square areas, which are line up on the same plane in the same direction [99]. The electrodes should be of the same length and width as that specimen. Pressure is applied to the specimen in a perpendicular plane. The capacitance is then measured using an L.C.R. meter, with an electric field applied over the thickness of the specimen.


**Measurement technique:**


The technique involved in measuring electrical permittivity is somewhat more complicated than measuring electrical resistivity because of the introduction of an interfacial capacitance at each electrode. In the case of sandwiched electrodes, the capacitance is perpendicular to the plane of the cement composite, and in the case of coplanar electrodes, the capacitance is in a plane with cement composite. The equations for the capacitors in series and in parallel are given below [104]:(6)$$\frac{1}{C_{m}}=\frac{1}{C}+\frac{2}{C_{i}}$$
(7)$$C_{m}=C+C_{o}$$
where *C_m_* is the measured capacitance, $C_{o}$ is the capacitance at *A* = 0, *C* is the volumetric capacitance of a cement composite and $C_{i}$ or $C_{c}$ is the interfacial capacitance. The volumetric capacitance is given by
(8)$$C=\frac{\xi_{o}kA}{l}$$
where $\xi_{o}$ is the permittivity of free space (8.85 × 10^−12^ F/m), *l* is the inter-electrode distance, *A* is the area of the specimen in the plane perpendicular to the direction of capacitance measurement and *k* is the relative permittivity of the specimen [103,105]. Therefore (6) and (7) become
(9)$$\frac{1}{C_{m}}=\frac{l}{\xi_{o}kA}+\frac{2}{C_{i}}$$
(10)$$C_{m}=\left(\xi_{o}kA\right)l+CC_{o}.\;$$

*C_i_* is replaced by $C_{o}$ as the interfacial capacitances are parallel to each other. Figure 8 represents a graphical plot between *1*/*C* and *l* where the slope ($\frac{\xi_{o}k}{l})$ can be determined. The intercept on the vertical axis at *l* = 0 equals 2/*C_c_*.

The parallel plate capacitor is easier to implement on a structure than the series method.


**Sensitivity measurement:**


The sensitivity measurement for capacitance is similar to the piezoresistivity measurement, and the fractional change in capacitance is derived from (8).

Since $C=\frac{\xi_{o}kA}{l},$
(11)$$\frac{\Delta C}{C}=\frac{\Delta k}{k}+\frac{\Delta A}{A}-\frac{\Delta l}{l}=\frac{\Delta k}{k}-\frac{\Delta l}{l}\left(1+2µ\right)$$
where µ is the Poisson ratio, $\frac{\Delta C}{C}$ is the fractional change in capacitance, Δk is the change in relative permittivity and Δl and ΔA represent the change in length and the change in area. From the above equation, both the change in *k* and the change in *l* contribute to the change in *C* [90]. Table 11 shows the conditions under which the capacitance changes. The negative sign implies compressive strain, and the positive sign indicates tensile strain. In condition 1, if $\frac{\Delta k}{k}$ is negative and $\frac{\Delta l}{l}$ is positive or vice versa, then the fractional change in capacitance is said to be negative, indicating a compressive strain. If both values are positive, the positive value indicates a tensile strain.

Conditions 2 and 3 are limited due to the negligible value of $\frac{\Delta k}{k}$, mainly due to the dimensional changes. This results in low sensing effectiveness. Condition 1 is preferred for high sensitivity [90].

### 3.3. Influencing Parameters


**Loading condition:**


The sensitivity parameter varies for different loading conditions. Figure 9 shows the graphical representation of load vs fractional change in resistance (F.C.R.). Three loading conditions were conducted in past literature: compression, tension and flexural.

In the case of compressive load, the graph is divided into three zones, as represented in Figure 9. Zone A represents the elastic regime; a conductive path is formed due to the increase in the tunnelling effect, in which on the removal of the load, the F.C.R. becomes 0. Zone B is formed after the elastic regime, where the micro-cracks start to originate inside the composite and where the conductive networks are reconstructed and form a balance stage. As loading continues, the cracks are propagated, and the conductive network breaks down, resulting in an increase in resistivity.

In the case of tension, the resistivity behaviour contrasts with the behaviour under compressive load, where the resistivity varies. On application of tensile loading conditions, the filler material separates and loses contact with itself, resulting in the breakdown of the network. Electrical resistivity increases with the increase of tensile stress, with a decrease in the tunnelling effect.

In the case of flexural behaviour, the resistivity behaviour follows the pattern of both compression and tension. On application of bending, the top region of the specimen undergoes compressive behaviour, and the bottom phase undergoes tensile behaviour, as shown in Figure 9.


**Curing age:**


The curing period is one of the important parameters that affect conductivity. On increasing the curing age, the hydration rate increases, which increases the hydration product, making the concrete member dense. The hydration product gets trapped in the pores, thus, limiting the formation of the conductive network [107].

Therefore, the electrical resistivity increases with longer curing. Figure 10 presents the differing resistivities of different materials at different curing times [108].


**Dispersion of filler material:**


The dispersion of conductive filler in the binder material plays an important role in the formation of a conductive network. If the filler material is not dispersed properly in the matrix medium, the matrix fails to hold the filler and results in low electrical and mechanical properties. The factors that influence the conductive fillers are the morphology and geometrical shape of the filler material, the dispersion medium, the surface features of filler and the dispersion method [109].

Two types of dispersion methods are adopted, physical, which includes ultrasonication, ball milling and shear milling [110,111,112], and chemical, like covalent and non-covalent fictionalization and plasma methods [113,114].


**Other influencing factors:**


In addition to the above influencing parameters, other factors such as temperature, water-cement ratio and freeze-and-thaw effects also have considerable effects on the sensing performance, and further research is needed on the above-mentioned topics.

## 4. Electrochemical Principle

An electrochemical reaction can be performed by placing two conducting materials (electrodes) into the cement composite (electrolyte) and connecting them electrically. The flow of current takes place through two reactions: electrons in the form of electrodes and ion carriers in the electrolyte. By measuring this current flow, microstructure, hydration, and several properties of the cement-based materials can be studied. AC impedance spectroscopy is one of the emerging techniques used based on the electrochemical principle to detect the behaviour of the composite member [104].

### 4.1. Alternate Current-Impedance Spectroscopy

AC impedance spectroscopy is based on the electrochemical principle in which on applying the voltage to a composite material through a proper mode of a channel, the mineral and chemical reactions in the composite start to respond to the applied parameter, resulting in the determination of the behaviour of the composite such as its microstructural characterization and structural performance [115,116,117,118]. This technique provides promising information on pore structure study, the hydration rate of cement-based materials, corrosion and permeability studies.

The Nyquist plot is a graphical chart representing the real and imaginary terms of impedance parameters that gives accurate information regarding the cement-based composite materials. Figure 11 presents a Nyquist plot that contains different circles and lines that represent the different frequency ranges [119,120]. The frequency arc denotes the bulk material effects and the polarization effect of the electrode and specimen, respectively [104].

The large diametric arc is treated as the low-frequency line. As the frequency range reduces to 10^−6^ Hz, a complete low-frequency arc is obtained, and a high-frequency arc is obtained by varying the frequency ranges and the geometry of the samples [119,120].

In order to obtain an accurate measurement, an equivalent circuit model is used. Appropriate equivalent circuit models can give information on microstructural characterization and structural health [121,122,123,124,125,126,127]. Figure 12 shows the experimental setup for AC impedance analysis, where the impedance is connected to the specimen through a circuit system.

### 4.2. Equivalent Circuit Model

Cement composite consists of a solid-liquid gel that comprises different chemical and mineral materials and has different electrical properties. On applying an electrical voltage, a complex electrochemical system is formed, and it is determined through either a parallel or series connection.

Many researchers have proposed several models for determining the behaviour of the cement composite. The circuit model consists of a resistor, capacitor and inductor as main parameters and a constant phase angle (CPE) that is treated as a distributing parameter in order to avoid the complications that are caused due to the rough solid–electrode interface [128,129,130]. Different electrical models have been established by researchers in order to determine the microstructural behaviour of the composite and the hydration rate of the material, analyse the effect of conductive filler in the medium, chloride migration, etc. The models represented in Figure 13, Figure 14, Figure 15 and Figure 16 show different electrical circuit models proposed by the researchers. This method provides an accurate result.

## 5. Applications

In practical health-monitoring applications, self-sensing materials are used in several structural features including beams, columns, bridges, etc. The following are the methods adopted for the implementation of self-sensing technology in structural applications [131,132,133,134]:In the bulk form, the structure is developed entirely using self-sensing materials, which both satisfies the structural health monitoring parameter and also strengthens the structure. It is easy to construct but economically expensive.Coated type—the non-conductive material is coated with a conductive or self-sensing medium. It provides both strength and sensitivity to the composite.Sandwiched type—involves covering the composite on the top and bottom surfaces using the conductive medium.Bonded type—The self-sensing sensors are attached over the surface of the composite.Embedded type—performed by inserting a self-sensing sensor inside the concrete composite. The sensors are typically as small as or slightly larger than the size of the conventional coarse aggregates.

In the transportation infrastructure, self-sensing techniques are implemented in pavements, roads, bridges, railway tracks, etc. The sensors collect information regarding vehicle speeds and flow rates, traffic density, moving weights, etc. [135,136,137].

## 6. Discussion

This paper provides an overview of the principles involved in the self-sensing technique. The researchers used different approaches to apply the self-sensing techniques, and the following conclusions are obtained from the study.

Electromechanical principle—the piezoresistive and piezopermittivity techniques were the focus. The filler material plays a major role in the conductivity and the piezoresistivity. The filler should not be less or more than the percolating value. The conducting mechanism depends on the tunnelling theory, where the electron transfer occurs when the filler particle gets overlapped. The four-probe method is more suitable for measuring the sensitivity parameters, but there are some complications in measuring due to the instrumental setup, whereas in the piezopermittivity method, the filler material is unnecessary for the conducting mechanism. The electrodes are replaced by a dielectric medium, the measuring technique becomes easier and the method becomes more economical due to the absence of conducting fillers. However, this technique fails to give an accurate sensing result.Electrochemical principle-based method—focused on AC impedance spectroscopy techniques. Various electrical circuit models were established to determine the pore structure, fibre orientation, corrosion monitoring and chloride migration. The recently developed method gives more accurate results for the sensing parameters.

Even though the electromechanical-based sensing techniques have several applications, the conductive filler material used for sensing is somewhat uneconomical in practical applications, and the results are not accurate. AC impedance spectroscopy provides better results and is also more economical than the above method.

## 7. Scope for Future Work

More research has to be focused on the AC impedance spectroscopy technique in practice. A suitable testing device has to be developed for easy implementation in practical applications.

## Figures and Tables

**Figure 1 materials-15-03831-f001:**
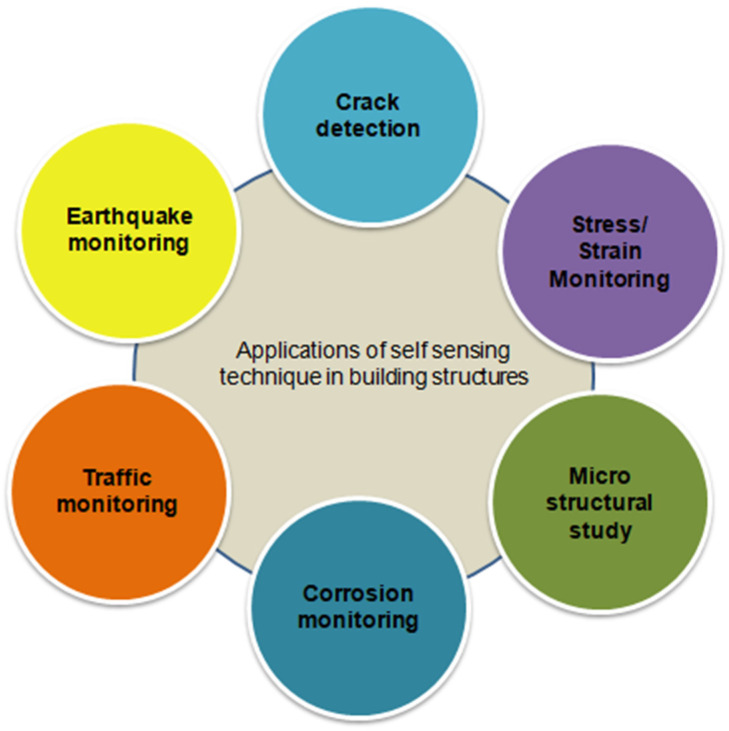
Applications of self-sensing in building elements.

**Figure 2 materials-15-03831-f002:**
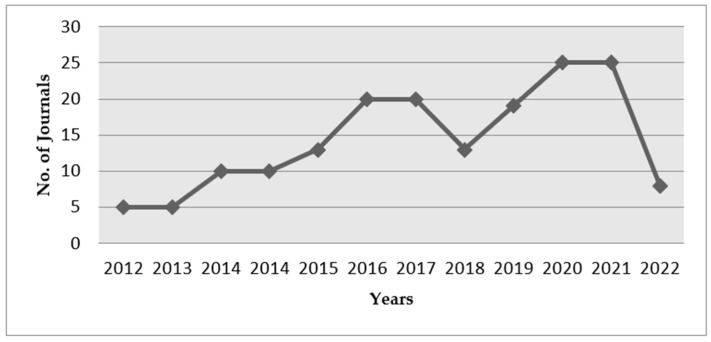
Past literature works (Scopus Indexed) published on self-sensing.

**Figure 3 materials-15-03831-f003:**
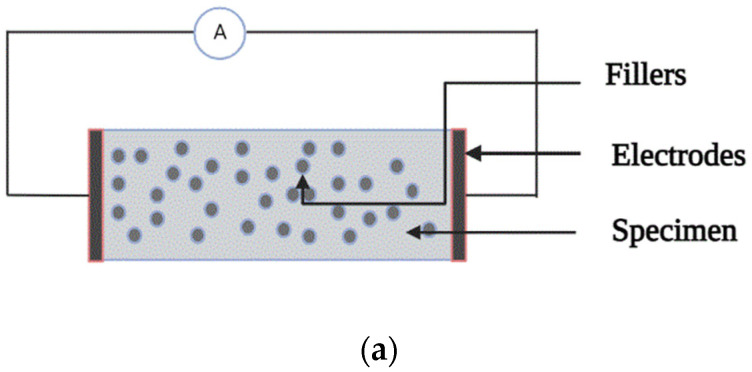
(**a**) Two-probe specimen setup. (**b**) Piezoresistive behaviour of the specimen under the loading condition.

**Figure 4 materials-15-03831-f004:**
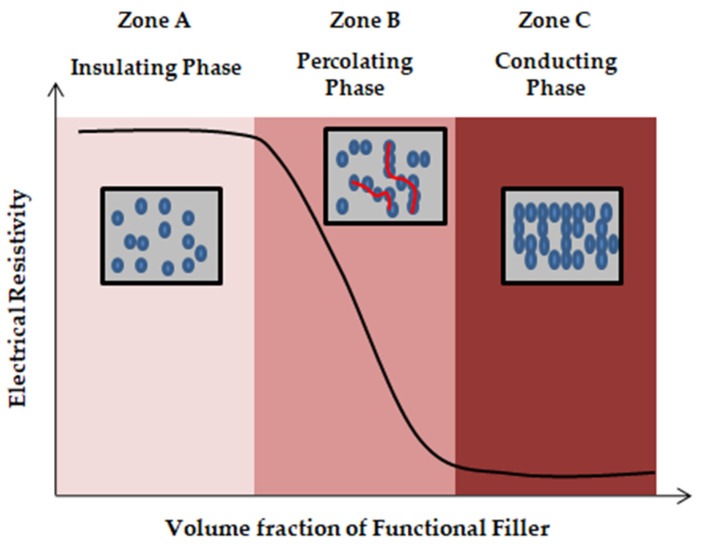
Percolation graph.

**Figure 5 materials-15-03831-f005:**
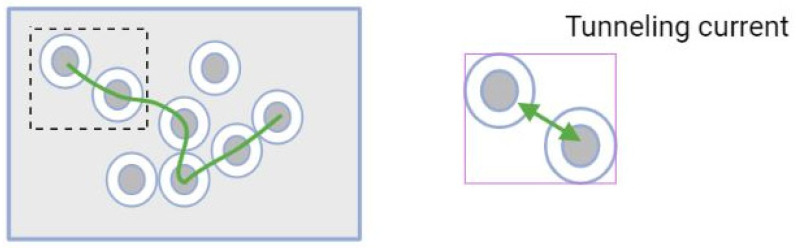
Thetunnelling effect.

**Figure 6 materials-15-03831-f006:**
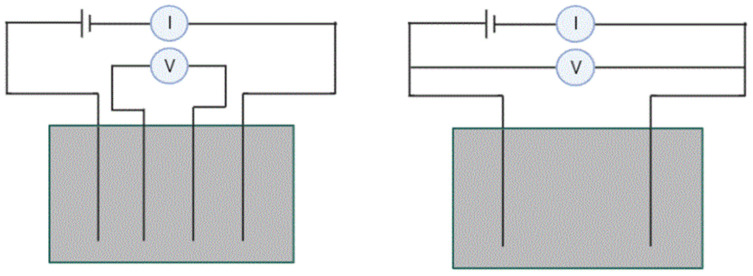
Four-probe and two-probe configurationsin concrete.

**Figure 7 materials-15-03831-f007:**
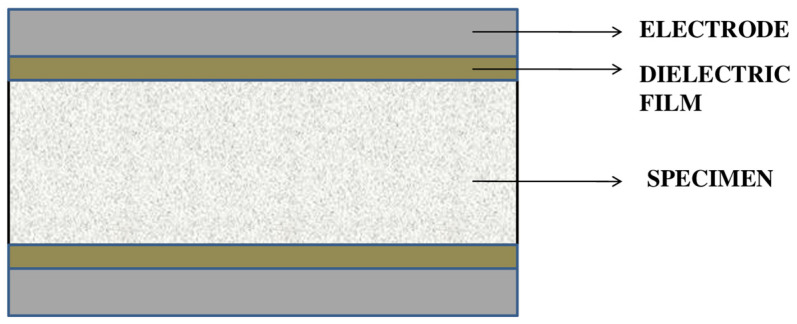
The specimen setup for capacitance measurement.

**Figure 8 materials-15-03831-f008:**
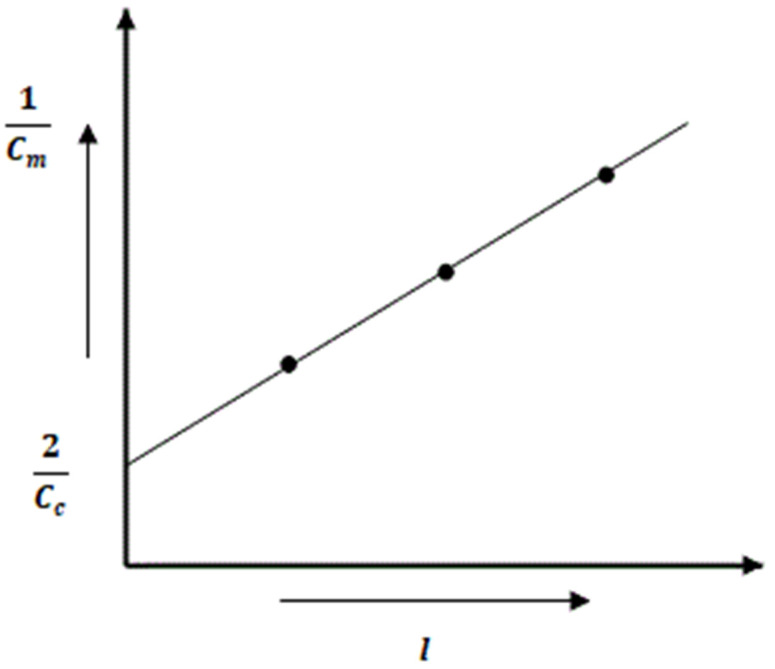
Graphical representation for determining ($\frac{\xi_{o}k}{l})$.

**Figure 9 materials-15-03831-f009:**
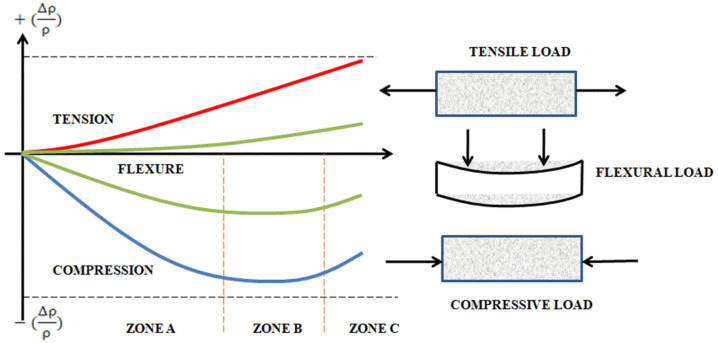
Graphical representation of fractional changes in resistivity for different loading conditions.

**Figure 10 materials-15-03831-f010:**
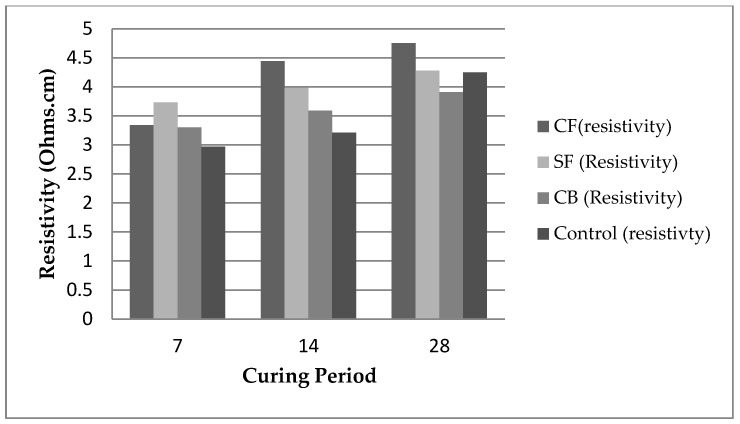
Curing effects for different fibres at different curing ages.

**Figure 11 materials-15-03831-f011:**
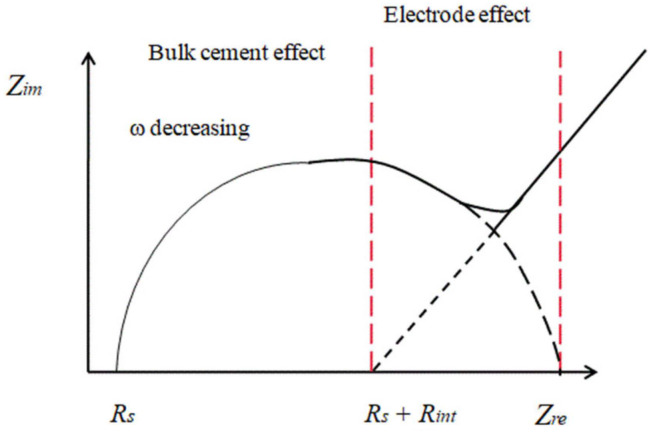
Nyquist curve.

**Figure 12 materials-15-03831-f012:**
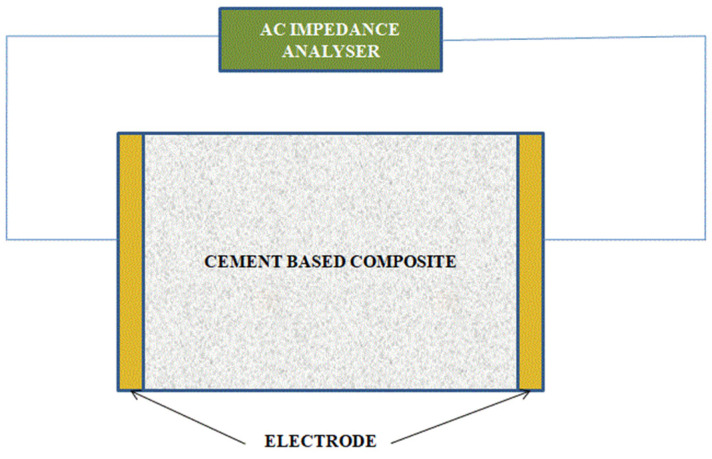
The specimen setup for AC impedance analysis.

**Figure 13 materials-15-03831-f013:**
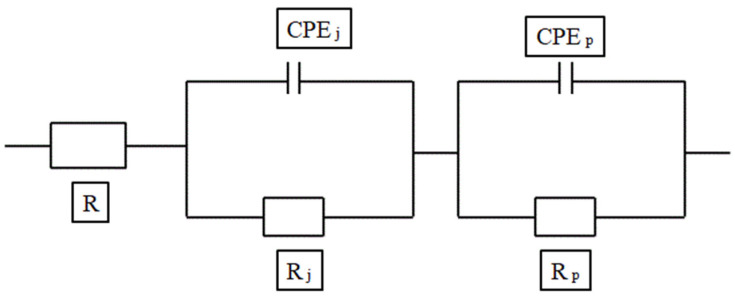
The circuit model for microstructural analysis.

**Figure 14 materials-15-03831-f014:**
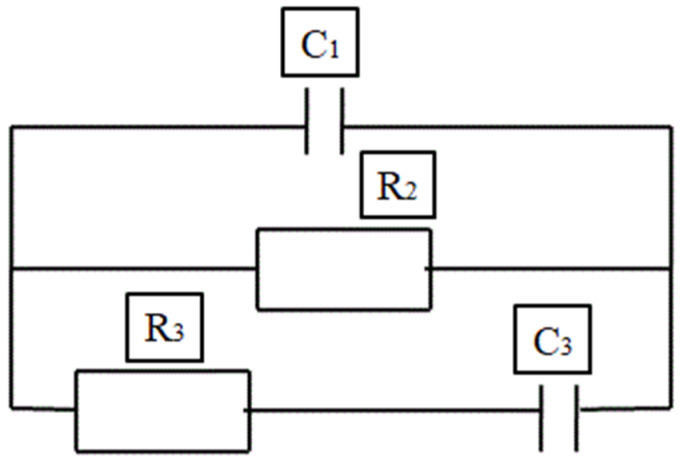
The circuit model for the hydration study.

**Figure 15 materials-15-03831-f015:**
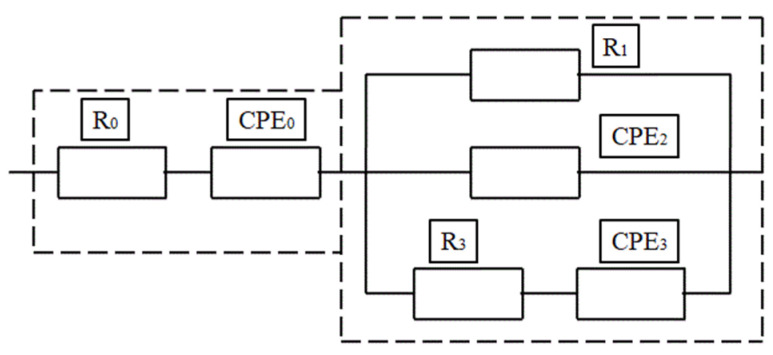
The circuit model for the analysis of the effect of the conductive medium in the composite.

**Figure 16 materials-15-03831-f016:**
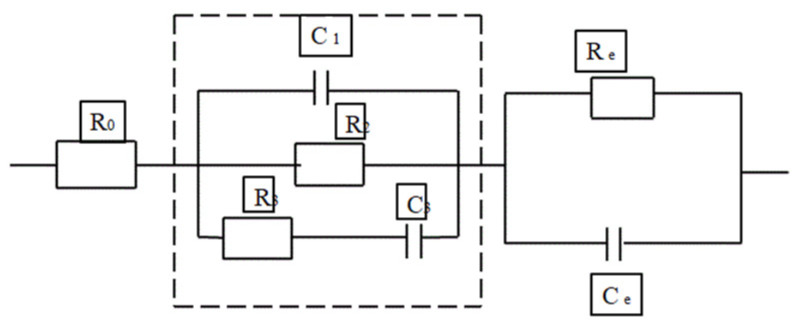
The circuit model for the analysis of chloride migration.

**Table 1 materials-15-03831-t001:** Conventional Sensors Used in Building Aspects.

Sensor Type	Application	Limitations
Accelerometer [6,7]	Measures the motion and vibration of a structure that is exposed to dynamic loads	Low durability and non-intrinsic
Extensometer [6,7]	Measures the elongation of material subjected to stress	Vulnerable and less sensitivity
Strain gauge [8]	Determines stresses in the structure by responding to the changes in dimensions due to creep, crack, temperature change, moisture change, etc.	Low sensing, worse durability and non-intrinsic
Piezoelectric sensors [9,10,11]	Measure impedance-based damage detection, guided wave damage detection, and structural dynamics applications in a structure	More AUD is required and non-intrinsic
Optical fibre sensors [12,13]	Measure the strain, temperature, and pressure in a structure	Vulnerable and non-intrinsic
Wireless smart sensors [14]	Detect, locate, and assess structural damages in a structure	Complication in implementing
Vision-based displacement measurement system [15]	Used for sensing the displacements in a structure	Not accurate and non-intrinsic
Shape Memory Alloy [16]	Used in building materials to withstand varied thermal conditions by gaining its original properties	More AUD is required and non-intrinsic

**Table 2 materials-15-03831-t002:** Properties adopted for different methods of electromechanical principle.

Condition	Piezoresistivity	Piezopermittivity	Piezoelectricity
**Mechanism**	Change in electrical resistivity on the application of external stimuli	Change in capacitance on subjected to external stimuli	Change in Electric field, on subjected to external stimuli
**Materials**	Conductive filler (steel fibre, nano nickel particles, carbon fibre, carbon nanotube, functionalized graphene groups) and non-conductive matrix (cement-based composites, alkali-based materials, etc.)	Composite (fibre reinforced polymer matrix, concrete), dielectric film and electrodes (copper, aluminium or steel)	Conductive filler (steel fibre, nano nickel particles, carbon fibre, carbon nanotube, functionalized graphene groups) and non-conductive matrix (cement-based composites, alkali-based materials, etc.)
**Dominant factor**	Current and Voltage	Frequency	Voltage

**Table 4 materials-15-03831-t004:** Influencing parameters of the percolation threshold.

Parameter	Formula	Description
Filler geometry	$V_{f_{-Sphere}}=\frac{\left(\pi D^{3}\right)}{6{\left(D+D_{IP}\right)}^{3}}$	D refers to particle size; $D_{IP}$ is interparticle distance of the filler [45].
	$V_{f_{-Planar}}=\frac{\left(2\pi tD^{2}\right)}{{\left(D+D_{IP}\right)}^{3}}$	
	$V_{f_{-3D}}=\frac{\left(27\pi tD^{2}\right)}{4{\left(D+D_{IP}\right)}^{3}}$	
Filler and matrix properties	$f_{c}\sim \;0.16\;\left(\frac{R_{1}}{R_{2}}\sim 1\right)\;$	f = volume fraction of minor phase, $f_{c}$ = percolation threshold,
	$f_{c}<0.16\;\left(\frac{R_{1}}{R_{2}}\gg 1\right)\;$	R_1_ = particle size of the major filler, R_2_ = particle size of the minor filler. Spherical filler in a homogeneous composite with random orientation [49]
	$f_{c}\ll \;0.16\;$	Ellipsoidal filler in an isotropic composite with random orientation [49]
Filler concentration	$\delta =\frac{\delta_{f}{\left(\varphi -\varphi_{c}\right)}^{\gamma }}{\left(1-\varphi_{c}\right)}$ (i) $\left(\varphi <\varphi_{c}\right)$ (ii) $\left(\varphi \;\sim \;\varphi_{c}\right)$ (iii) $\left(\varphi >\varphi_{c}\right)$	δ = electrical conductivity of the material, δ_f_ = conductivity of the filler material, φ = filler concentration, φ_c_ = percolation threshold, γ = universal critical exponential [44]. (i) Insulation Zone (ii) Percolation Zone (iii) Conducting Zone

**Table 5 materials-15-03831-t005:** Percolation thresholds for different filler materials with respect to their matrix.

Filler	Matrix	% of Fibres	Percolation Threshold (%)	References
Carbon black	Cementitious material	0.2–0.5	7.22–11.39	[49]
Expanded graphite	High Density Polyethylene	0.1	4.46	[50]
Graphite	Epoxy	0.5	1.13	[51]
Graphite	Poly(styrene-methyl methacrylate)	0.5, 1	0.878	[52]
Expanded graphite	Polymethylmethacrylate	1	0.529	[52]
Graphite nanoplatelets	Polypropylene	-	0.67	[53]
Graphite nanoplatelets	Epoxy	0.2, 0.4, 0.6	0.5	[54]
Graphite nanoplatelets	Polymer composite	0.5	0.52	[55]
Carbon fibres	polymer matrix	1, 1.5	0.9	[56]
MWCNT	Cementitious material	0.5, 1.15	1.00	[56]
MWCNT	Cementitious material	1	1.15	[56]
MWCNT	Cementitious material	0.3–0.6	0.35–0.7	[56]

**Table 6 materials-15-03831-t006:** Resistivity of materials according to contact mode.

Matrix	Fibre (%)	Method	Electrode Type	Current Type	Resistivity (Ω·cm) × 10^3^	References
Alkali activated blast furnace slag	Carbon fibre (0.58)	Four-probe method	Silver paint wrapped with copper wire	DC	9.956	[61]
E.C.C.	Carbon fibre (1)	Surface electrodes	An electrode made up of copper plate	AC	7.5	[62]
ECC	CNT (0.5)	Surface electrodes	An electrode made up of copper plate	AC	84.5	[62]
E.C.C.	Carbon black (0.01)	Surface electrodes	An electrode made up of copper plate	AC	97.34	[62]
UHPC	Steel fibre (2)	Two-probe method	-	AC	420	[68]
Concrete	MWCNTs (0.05)	Four-probe method	An electrode made up of copper plate	DC	181	[68]

**Table 7 materials-15-03831-t007:** The resistances for the different types of circuit current.

Circuit Type	Ohm’s Law	Description
Direct current circuit	V=IR	*V* refers to the voltage (V), *I* is the intensity of the current (A) and *R* is the electrical resistance (Ω) [68].
Alternating current circuit	$Z=\frac{I}{R}$	*Z* is the impedance (including resistance and reactants), which refers to the total opposition of the current flow [70].

**Table 8 materials-15-03831-t008:** The conditions adopted for determining sensitivity.

Condition	F.C.R.	Gauge Factor	Sensitivity Criterion
$\frac{\delta \rho }{\rho }\gg \frac{\delta l}{l}$	$\frac{\delta R}{R}=\frac{\delta \rho }{\rho }$	$\frac{\left(\frac{\delta R}{R}\right)}{\frac{\delta l}{l}}=\frac{\left(\frac{\delta \rho }{\rho }\right)}{\frac{\rho l}{l}}$	In case 1, the gauge factor is dictated by the change in resistivity (*δρ*/*ρ*) and has a magnitude that depends on the piezoresistivity of the material, and it is not limited.
$\frac{\delta \rho }{\rho }\sim \frac{\delta l}{l}$	$\frac{\delta R}{R}=\left(\frac{\delta l}{l}\right)\left(2+2µ\right)$	$\frac{\delta R}{R}=\left(2+2µ\right)$	In case 2, the maximum value of µ is 0.5, so the maximum value that the G.F. can obtain is 2, which is low.
$\frac{\delta \rho }{\rho }\ll \frac{\delta l}{l}$	$\frac{\delta R}{R}=\left(\frac{\delta l}{l}\right)\left(1+2µ\right)$	$\frac{\delta R}{R}=\left(1+2µ\right)$	In case 3, the maximum G.F. that can be obtained is 3, which is low.

Based on the above three conditions, case 2 and case 3 are limited due to low G.F. values; thus, case 1 is assumed to be suitable for obtaining high sensitivity.

**Table 9 materials-15-03831-t009:** Sensing parameters for different fibre proportions.

Type of Filler	Type of Matrix	Percentage of Filler Material (%)	Sensitivity Properties			References
			F.C.R.	Gauge Factor	Resistivity (Ohms·cm)	
Steel Fiber	Cementitious matrix	0.5	-	87.26	102.86	[81,82,83]
		1	-	155.99	21.43	
		1.5	-	164.24	17.13	
		2	-	156.45	11.39	
	Concrete	20	0.194	1.78	-	
		40	0.13	4.68	-	
		60	0.122	0.77	-	
	Cement mortar	Lengthy twisted (1.5)		138.09	55.54	
		Lengthy smooth (1.5)	-	99.85	109.06	
		Lengthy hooked (1.5)	-	88.5	175.03	
		Medium twisted (1.5)	-	139.68	113.58	
		Medium smooth (1.5)	-	99.7	352.11	
		Short smooth (1.5)	-	52.9	628.97	
Carbon Nanotube	Cement paste	0.2	0.02	-	-	[84,85,86]
		0.3	0.03	-	-	
	Cement paste	0.6	-	1	1	
		0.7	-	50	50	
		1.2	-	2	2	
	Concrete	0.25	20	-	-	
		0.5	25	-	-	
Carbon Fiber	Concrete	0.5	12.5	-	-	[87,88]
		1	11	-		
	Cement paste	0.5		405.3		
	Cement paste	0.1	13			
		0.5	3			
		1	2			
	Concrete	0.5	0.37			
		2	1.01			
		3	1.32			
C.F., C.N.T.	Cement paste	0.1, 0.5		160.3	25	[88]
S.F., C.N.T	Concrete	2, 0.5	0.236	67.8		[89]
S.F, CB	Concrete	20, 1 (kg/m^3^)	0.323	1.08	-	[90]
S.F, CB, CF	concrete	60, 1, 2 (kg/m^3^)	0.169	1.55		[90]

**Table 10 materials-15-03831-t010:** Permittivity and resistivity values for different materials.

Conducting Type	Material	Permittivity (F/m)
Conducting	CFRP	1.6 × 10^3^
Conducting	Copper	2.4 × 10^3^
Conducting	Carbon fibre	4.0 × 10^3^
Non-Conducting	Cement paste	28
Non-Conducting	Mortar	13.2
Non-Conducting	Concrete	11.9

**Table 11 materials-15-03831-t011:** Conditions adopted for determining sensitivity nature.

Condition	Sensing Effectiveness
$\frac{\Delta k}{k}\gg \frac{\Delta l}{l}$	$\frac{\frac{\Delta C}{C}}{\frac{\Delta l}{l}}=\frac{\frac{\Delta k}{k}}{\frac{\Delta l}{l}}$
$\frac{\Delta k}{k}\sim \frac{\Delta l}{l}$	$\frac{\frac{\Delta C}{C}}{\frac{\Delta l}{l}}=-2µ$
$\frac{\Delta k}{k}\ll \frac{\Delta l}{l}$	$\frac{\frac{\Delta C}{C}}{\frac{\Delta l}{l}}=-\left(1+2µ\right)$

## Data Availability

Not applicable.

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
