# Peer review of "A Review on Principles, Theories and Materials for Self Sensing Concrete for Structural Applications"

_materials, 2022, doi:10.3390/ma15113831_

Round 1
Reviewer 1 Report
The review is devoted to an overview of self-sensing concrete including, the principles such as piezoresistivity and piezopermittivity, the theories - tunneling effect, percolation threshold, and electrical circuit theories, the materials used, methods adopted, and sensing parameters. It is actual and interesting problem nowadays.
History of development of the Structural Health Monitoring sensors is described in the review. Since 19th century, when the strain gauge was the first sensing device introduced to monitor the displacement and stress, followed by accelerometer and extensometer sensors till later, when smart sensors like piezoelectric sensors, fiber optical grating, wireless sensors and shape memory alloys came into applications. Due to their drawbacks such as poor durability, short span, low sensitivity, and low compatibility, the research and development have recently introduced intrinsic self-sensing concrete, which can satisfy both the strength parameter and also structural health monitoring purposes.
Comments and questions:
- The phrase on page 2 isn’t finished.
- There is a print mistake in formula Nr. 2 (or description), page 10
- There is a lack of references to Tables 6 and 7.
- What does mean point by variable R. in formula Nr.3?
- It’s desired the description for formula Nr. 4 would be in one style.
- What does mean variable in last paragraph on page Nr. 12, as well in the description Case 1 in Table 8?
- Could you explain, why the Gauge factor and Resistivity (Ohms.cm) are fully identical in table 9.
- There is a lack of references to Table 9 and an explanation of the information shown in this table.
- There is a lack of measurement units for the parameters in the several Tables. You need to check all tables.
- There are some problems with page numbering beginning from the 17 page and with lines numbering from the beginning of paper.
- What is Cm and C0 in equation 6?
- There is a lack of references to Tables 11 and figure 8 with an explanation.
- There is a lack of references to figures 11 and 12 with an explanation and a wrong reference to Figure 10 on page 20.
- There is a lack of references to the paper Nr. 149.
- There a lack of Conclusions in the paper.
- There is a lack of numerical evaluation of the described methods.
- Is it possible to develop a universal method for a definite type of structure element or the monitoring passport should be developed for each structure element before monitoring?
Reviewer 2 Report
Dear author,
This article extensively reviewed the recent advancements and outlook on self-sensing concrete and its implementation. This article focused on the principles such as piezoresistivity and piezopermittivity, the theories of tunneling effect, percolation threshold, electrical circuit theories, materials used, methods adopted, and sensing parameters. The review article is related to smart materials and is beneficial to structural health monitoring research areas.
Please consider the following remarks.
- The two points should be highlighted in the abstract: What is the main objective and necessity of this review? What is the outcome of this review?
- Figure 1 has not been cited in the text. Please check.
- It is better to add a caption for the x and y-axis in Figure 2.
- More references could be included in the Introduction section.
- Section 3.1. It is better to highlight why piezoresistivity is the most widely used method in self-sensing concrete.
- Table 3. The density of steel fibers should be included in Table 3.
- The clarity of figure 4 should be improved.
- It should be explained in more detail how the Filler Orientation, Filler and matrix properties, and Filler concentration influence the percolation threshold.
- Page 15. Please explain why the piezopermittivity is more advantageous than piezoresistivity. This advantage should be highlighted in the manuscript.
- Figure 10 can be improved.
- The equation is part of the text of the article. The equation must be followed by a punctuation mark, such as a comma or period.
- The word "where" after the formula (7) is written with a lowercase letter, and after the formula (8) is written with a capital letter. Template's style MDPI_3.2_text_no_ident is not applied. Formulas are formatted not in accordance with the template. These technical details create a negative impression of the article even with good content.
- References are not formatted correctly. Using the free Mendeley reference manager (or similar software) is recommended. Use MDPI citation style, available for installing to Mendeley software by entering the link http://www.zotero.org/styles/multidisciplinary-digital-publishing-institute. Be sure that you choose the right citation style in MS Word.
- Please recommend the scope for future work.
Reviewer 3 Report
I would strongly suggest that the authors have this paper reviewed and corrected for proper translation into English. I started to list the errors, but halfway through the abstract, I decided there were going to be far too many to address individually. The quality of the writing is still insufficient for a published journal, in my opinion. Open Access journal should not be the synonym for low-quality work. Below I list some of the errors. I am happy to review this manuscript again if the quality of this manuscript is significantly improved.
Page 2. Delete the duplicate of "To overcome the above issues, the research and development..."
Page 3. Figure 1. There are two "Earthquake monitoring". What are the differences between them?
Page 3. Figure 2. Number of journals published? What does this mean?
Page 9. Table 5. %of fibers?
Round 2
Reviewer 3 Report
This manuscript is well-written and well-organized. This manuscript is recommended for being accepted in present form.